# Identification of a Maturation Plasma Cell Index through a Highly Sensitive Droplet Digital PCR Assay Gene Expression Signature Validation in Newly Diagnosed Multiple Myeloma Patients

**DOI:** 10.3390/ijms232012450

**Published:** 2022-10-18

**Authors:** Marina Martello, Vincenza Solli, Rosalinda Termini, Ajsi Kanapari, Daniel Remondini, Enrica Borsi, Andrea Poletti, Silvia Armuzzi, Barbara Taurisano, Ilaria Vigliotta, Gaia Mazzocchetti, Elena Zamagni, Alessandra Merlotti, Paola Tacchetti, Lucia Pantani, Serena Rocchi, Ilaria Rizzello, Katia Mancuso, Michele Cavo, Carolina Terragna

**Affiliations:** 1IRCCS Azienda Ospedaliero-Universitaria di Bologna, Istituto di Ematologia “Seràgnoli”, 40138 Bologna, Italy; 2Department of Experimental, Diagnostic and Specialty Medicine, University of Bologna, 40138 Bologna, Italy; 3Clinica Universidad de Navarra, Centro de Investigación Médica Aplicada (CIMA), Instituto de Investigacion Sanitaria de Navarra (IDISNA), CCUN, CIBER-ONC Numbers CB16/12/00369, CB16/12/00489, 31001 Pamplona, Spain; 4Department of Physics and Astronomy, DIFA—University of Bologna, 40126 Bologna, Italy

**Keywords:** multiple myeloma, hedgehog signaling, self-renewal, digital PCR, plasma cell maturation, differentiation stages, gene expression

## Abstract

DNA microarrays and RNA-based sequencing approaches are considered important discovery tools in clinical medicine. However, cross-platform reproducibility studies undertaken so far have highlighted that microarrays are not able to accurately measure gene expression, particularly when they are expressed at low levels. Here, we consider the employment of a digital PCR assay (ddPCR) to validate a gene signature previously identified by gene expression profile. This signature included ten Hedgehog (HH) pathways’ genes able to stratify multiple myeloma (MM) patients according to their self-renewal status. Results show that the designed assay is able to validate gene expression data, both in a retrospective as well as in a prospective cohort. In addition, the plasma cells’ differentiation status determined by ddPCR was further confirmed by other techniques, such as flow cytometry, allowing the identification of patients with immature plasma cells’ phenotype (i.e., expressing CD19+/CD81+ markers) upregulating HH genes, as compared to others, whose plasma cells lose the expression of these markers and were more differentiated. To our knowledge, this is the first technical report of gene expression data validation by ddPCR instead of classical qPCR. This approach permitted the identification of a Maturation Index through the integration of molecular and phenotypic data, able to possibly define upfront the differentiation status of MM patients that would be clinically relevant in the future.

## 1. Introduction

Heterogeneity is a hallmark of multiple myeloma (MM) genome and transcriptome, and several molecular tests have been employed in the last few years in order to dissect patients’ molecular profiles, finally aiming at risk assessment and prognosis definition [1,2,3].

In this context, DNA microarrays and, more recently, RNA sequencing, can simultaneously measure the expression level of thousands of genes up to the entire transcriptome [4,5]. Such high-throughput expression profiling can be mainly used to compare the level of gene transcription in clinical setting in order to: (1) identify diagnostic or prognostic biomarkers; (2) classify diseases (e.g., tumors with different prognosis that are indistinguishable by microscopic examination); (3) monitor the response to therapy; and (4) understand the mechanisms involved in the genesis of disease processes [6]. For these reasons, DNA microarrays and RNA sequencing approaches are considered important discovery tools in clinical medicine. However, cross-platform reproducibility studies undertaken so far have highlighted that microarrays are not able to accurately measure gene expression, particularly when low [7,8,9,10]. 

In a previous work [11], we have demonstrated a differential expression of the Hedgehog (HH) pathway, a self-renewal signaling involved in tumorigenesis and maintenance of cancer cells, among newly diagnosed multiple myeloma (NDMM) patients. In particular, patients were stratified in two clusters according to a 10 HH genes signature (Ligands: SHH, DHH, IHH; Receptors: PTCH1, PTCH2; Intermediate: SUFU; Transcription factors: GLI1, GLI2, GLI3) expression profiles. Cluster 1 patients were characterized by an overall downregulation of the pathway, whereas Cluster 2 patients displayed marked overexpression of the pathway. Although several studies pinpoint the relevant role of self-renewal mechanisms in neoplastic diseases [12,13], overall, we observed that this pathway was expressed at low level in MM patients compared to normal samples [14]. Therefore, a robust technique should be employed in order to validate these data.

Droplet digital PCR assay (ddPCR) technology is a digital PCR method utilizing a water-oil emulsion droplet system. The creation of tens of thousands of droplets means that a single sample can generate tens of thousands of data points rather than a single result, bringing the power of statistical analysis inherent in digital PCR into practical application [15,16]. This technique requires the input of small sample quantities, which reduces costs and preserves precious samples. For all these reasons, ddPCR technology offers several advantages compared to real-time quantitative PCR (qPCR), including the attainment of absolute quantitative results, improved precision, reproducibility, accuracy, sensitivity, and great tolerance to PCR inhibitors. Moreover, nucleic acid quantitation is independent of reaction efficacy, providing greater ability to detect and accurately quantify low-abundance targets [17].

Here, we consider the employment of a ddPCR assay to validate a gene signature, named the 10 HH genes signature, previously identified by gene expression profile. Results show that the designed assay is able to validate gene expression data, both in a retrospective as well as in a prospective cohort. In addition, the plasma cells’ differentiation status determined by ddPCR could be further confirmed by other techniques, such as flow cytometry. To our knowledge, this is the first technical report of gene expression data validation by ddPCR instead of classical qPCR. The use of ddPCR permitted the identification of a Maturation Index capable of discriminating between patients with mature plasma cells (PCs) and patients with immature PCs.

## 2. Results

### 2.1. 10 HH Genes Were Expressed at Low Level across Different Cohorts of NDMM Patients

Since the current aim of the work was to validate the previously defined 10 HH genes signature, we evaluated which technique might best reproduce the benchmark results.

As we recently reported, the 10 HH genes are expressed at low level in MM patients [11]. To further confirm this, we compared gene expression data obtained from nine different cohorts of NDMM patients, downloaded from the GEO database (Appendix A), including a total of 3023 patients, to gene expression data obtained from 27 donor samples, with a focus on the 10 HH genes expression data (Figure 1). In order to avoid platform biases, only cohorts including at least 50 NDMM patients were considered, preferably profiled by Gene Chip Plus 2.0 (Affymetrix, Thermo Scientific). The comparisons confirmed that Hedgehog pathway genes were low when expressed in MM compared to donor samples. Indeed, the 10 HH genes are not included in the gene list commonly implicated in MM disease onset; in addition, their fold-change (FC) values fall within the −2; 2 FC interval (10 HH genes FC range: from −0.56 to 0.32; Table 1).

For this reason, in order to validate the signature, we considered ddPCR the best approach available, as it is able to detect low abundant targets more efficiently than conventional qPCR.

### 2.2. Selection of Suitable Housekeeping Gene

In order to properly perform the 10 HH genes quantification via ddPCR, the intrinsic expression values variability need to be normalized using a house-keeping (HK) gene.

To this end, the normalized expression (NE) values of 13 known HK genes (B2M, CDKN1A, CTBP1, GUSB, MTBP, PPIH, PTBP2, PUM1, SETBP1, TBP, TBPL1, TFRC, and YWHAZ) were extrapolated from our 126-gene expression array data, and three HK genes, whose expressions were quantitatively similar to those of the 10-HH genes signature (median NE values = 45.0), were selected in order to perform data normalization. In particular, a high (PUM1: 726.03 NE), an intermediate (PPIH: 145.08 EV), and a low (MTBP: 22.09) HK gene were identified (Figure 2). Of these three HK genes, MTBP was selected since its expression was most similar to that of the 10 HH gene signature (22.09 vs. 45.0 NE values) and, more importantly, because it is located out of chromosomal regions frequently affected by copy number variation (CNV) (i.e., on chr8q24.12).

### 2.3. Design and Set-Up of the 10 HH Genes Signature ddPCR Assay

In order to design a 10 HH gene signature ddPCR assay, we extrapolated from the Plus 2.0 gene chip array the exact genomic sequences of each HH gene’s probe (~500 bp each) (Appendix A). Custom ddPCR assays consist of two primers, forward and reverse, and an intermediate probe conjugated with FAM and HEX fluorophores, allowing the possibility ofall the assays being multiplexed in a unique cartridge.

Custom assays were first tested on a MM cell line, NCI-H929, which represented the ideal positive control of Hedgehog pathway expression [18]. Preliminary data were produced by employing four different bona fide human MM cell lines (MM1.S, KMS12BM, NCI-H929, and RPMI-8226), whose CEL file was downloaded from Array Gene Express and GEO database (E-GEOD-22759 and GSE53798, respectively). We first processed the gene expression profiles of cell lines as described in the Patients and Methods section in order to assign them to either Cluster 1 or Cluster 2, according to the HH-gene signature. As shown in Table 2, the signature was able to classify each cell line as belonging either to Cluster 1 (MM1.S and RPMI-8226) or to Cluster 2 (KMS12.BM and NCI-H929). Assays were initially performed as a single assay, and a non-template control (NTC) was included in the experiment for each gene. Then, we tested different target concentrations by amplifying 5, 25, and 50 ng of input material (cDNA). In order to verify the reproducibility between replicates and between different experiments, all three target concentrations were analyzed in replicates, two times in two independent experiments. Since low quantification results were obtained for some genes, particularly for SHH, IHH, and GLI2, we finally set the input cDNA material at 50 ng (the highest tested), the right amount required for the optimal target identification. Notably, good quantification results’ reproducibility between replicates and experiments was also observed (rho = 0.9966) (Figure 3).

### 2.4. Multiplexed ddPCR 10-HH Genes Assay: Signature Validation on Primary Patients’ Samples

The different assays were then multiplexed, and the cartridge was optimized by coupling genes with similar expression values in the same well (Table 3). To this end, SMO, SHH, PTCH1, IHH, DHH, and MTBP were conjugated with FAM fluorophore, whereas SUFU, GLI1, GLI2, GLI3, and PTCH2 were conjugated with HEX fluorophore.

A total of 126 patients was screened via the ddPCR assay: 39 derived from Cluster 1; 37 from Cluster 2; and 50 initially attributed to the test set. In order to normalize gene expression values, all the absolute quantification values were corrected by the MTBP gene expression. According to the ddPCR results, all patients were then assigned either to Cluster 1 or to Cluster 2. By comparing the median expression values of each gene of the signature, 6 out of 10 genes showed a significant difference between the two clusters of patients (*p* < 0.05). In particular, patients included in Cluster 2 displayed an overall activation of the pathway, accounted for by at least two ligands (DHH, IHH), two receptors (SMO, PTCH1), the intermediate (SUFU), and one transcription factor (GLI2) (Figure 4).

Notably, ddPCR results also made it possible to correctly assign patients, first included in the test set, to their own Cluster, previously defined according to the distance from either Cluster 1 or Cluster 2 in terms of gene expression values (Appendix A). In particular, 31 patients were assigned to Cluster 1 and 19 to Cluster 2; the comparison of gene distances between Clusters, as obtained both from gene expression data and from ddPCR data overall, showed a good correlation (*p* = 0.0001; r2 = 0.98).

Therefore, ddPCR quantification of the 10 HH genes signature has been shown to be able to correctly reproduce results previously obtained by gene expression profiles, identifying patients with different expressions of HH pathway.

### 2.5. Prospective Testing of the 10 HH Genes Signature by Integration of Molecular and Immunophenotypic Data

Data from literature have suggested that the Hedgehog pathway plays an important role in cancer cells’ replication, survival, and differentiation [18,19,20]. Accordingly, plasma cells might express the Hedgehog Pathway differently based on their differentiation status [11]. In order to prospectively validate the 10 HH genes signature by ddPCR, 22 NDMM patients were screened up front for an immunophenotypic profile, which included, among others, specific surface markers related to plasma cells differentiation (e.g., CD19, CD81). According to their expression, 10 of 22 patients were defined as CD19+/CD81+ (i.e., with immature/less differentiated PCs), whereas 12 out of 22 were defined as CD19-/CD81- (i.e., with mature/more differentiated PCs). Testing the 10 HH genes signature in this prospective cohort confirmed their differentiation status. Indeed, even though the genes’ absolute quantification values were at least 10-fold higher with respect to those obtained in the validation cohort, the comparison of the median expression values of the 10-HH genes in the two immunophenotypic groups highlight that 3 out of 10 genes were significantly different between the two groups (SMO, PTCH1, and GLI1; *p* < 0.05). Interestingly, patients with a CD19-/CD81- profile, characterized by a more differentiated PC status, displayed an overall activation of the Hedgehog pathway with respect to patients with an immature CD19+/CD81+ PCs profile (Table 4).

Therefore, the integration of molecular and immunophenotypic information permitted the fine description of the MM PCs’ differentiation status at diagnosis, allowing the definition of a PCs Maturation Index, possibly supporting the process of patients’ stratification.

## 3. Discussion

For many years, gene expression profiling via microarray has represented a high-resolution and valid technique that has enabled the identification of gene expression fluctuations under different experimental conditions [20,21,22]. Moreover, the extrapolation of a restricted list of genes, named “signature”, has allowed patients’ stratification in prognostic meaningful subgroups [23,24,25]. However, one of the pitfalls of this approach has been represented by the need to be validated by other techniques, which possibly should be more manageable and ready to use in daily clinical practice. From these perspectives, ddPCR technology might be considered the ideal validation approach, since it provides an absolute count of target DNA copies per input sample, without the need to run standard curves such as in the traditional quantitative real-time PCR [26,27]. Indeed, it permits the absolute quantification even of very low abundant targets, such as the 10 HH gene signature herein tested [14]. Moreover, it is faster, as compared to a conventional gene expression profile experiment, and results are available in a few hours (instead of in days).

The setup of a new custom ddPCR, such as the one presented in this work, has required the preliminary investigations of several experimental conditions on MM cell lines. For example, the choice of the MTBP gene as the housekeeping gene, to be employed for expression data normalization, has been mainly conditioned by three reasons, which overall have facilitated the normalization and the development of the assay: (1) MTBP gene expression was the most similar to those of the 10 HH genes; (2) in comparison to other low-expressed HK genes, such as PPIH (chr1p34.2) and PUM1 (chr1p35.2), MTBP is located in a chromosomal region (chr8q24.12) that is infrequently affected by copy number alterations; (3) after the set-up phase, we established that just one housekeeping gene should be maintained, in order to optimize the final cartridge layout to allow the analysis of one patient at a time [28]. In addition, the identification of the optimal input template, which is not entirely negligible. Indeed, the use of the highest possible amount of input material (i.e., 50 ng) has been considered necessary compared to 5 and 25 ng, since the HH genes have been shown to be low expressed in MM patients.

Recent published works suggested that the HH pathway plays an important role in cancer cells’ replication, survival, and differentiation [20] and, accordingly, plasma cells might express the Hedgehog Pathway differently based on their differentiation status [11]. Therefore, the possibility to define upfront the differentiation status of MM patients, according to the expression of these genes, will be clinically relevant in the near future. In order to validate the HH genes’ signature via ddPCR, we planned to test it both in a retrospective as well as a prospective cohort of MM patients.

Results on the retrospective cohort of patients demonstrated that the ddPCR assay has been able to reproduce the results obtained by gene expression profiles. Indeed, we showed that the HH pathway quantification by ddPCR correctly identified patients previously included in Cluster1 as effectively downregulating the HH pathway and, on the contrary, patients previously classified in Cluster2 were identified as overexpressing the pathway. Therefore, the 10 HH genes signature tested by ddPCR has been able to identify patients with an impaired self-renewal ability, ultimately correlated with a worse outcome.

To further confirm these data, the ddPCR assay was employed to evaluate the HH gene expression on a prospective cohort of patients. Although this patient population was relatively small, an immunophenotypic profile allowed the stratification of patients according to the presence of either mature or immature plasma cells according to the expression of CD19 and CD81 markers. As a general observation, the HH genes’ absolute quantification values were at least 10-fold higher compared to those obtained in the validation cohort; this is probably due to the better quality of input RNA, which was extracted just before the ddPCR experiment, as opposed to the validation set’s RNA, which was older (the quantity was always the same). Testing the 10 HH genes signature by ddPCR assay in this cohort of patients confirmed that patients with less differentiated PCs actually downregulated the HH pathway, as we previously observed through gene expression profiling.

This opens the possibility to integrate molecular and immunophenotypic data in a Maturation Index, which might be prospectively employed, in order to stratify patients according to the differentiation status of their PCs both at diagnosis and during the disease course, in order to evaluate whether therapy selective pressure might possibly change the PCs clone’s molecular and immunophenotypic make-up.

Finally, the 10 HH genes signatures’ ddPCR assay might represent a good example of validation and translation of gene expression profile data into a ready-to-use technique, which is possibly prospectively applicable. Validation on the retrospective cohort strongly supports its application in the near future. However, the development of a Maturation Index through molecular and immunophenotypic data integration needs to be further validated, even though these preliminary data suggested its relevance in order to stratify patients according to the differentiation status of their plasma cells.

To our knowledge, this represents the first technical report of a validation of gene expression data by ddPCR instead of classical real-time qPCR. In addition, it offers suggestions on how to develop a Maturation index that possibly defines the PCs’ differentiation status of MM patients upfront, information that would be clinically relevant in the near future.

## 4. Patients and Methods

### 4.1. Patients

For study purposes, the 10 HH genes signature ddPCR assay was validated both in a retrospective cohort of one hundred twenty-six patients (aged 18–65 years) with NDMM, as well as in an independent prospective cohort of twenty-two NDMM patients. Written informed consent was obtained from each patient. Molecular characteristics of the retrospective cohort have been previously reported [11], and patients were divided into: Cluster1 (Clu1), including 39 patients who under-expressed the 10 HH genes signature; Cluster 2 (Clu2), composed of 37 patients who overexpressed the pathway; and “test-set”, including 50 pts, whose affinity with one cluster or with the other has previously been estimated, according to the distance in terms of gene expression from either Clu1 or Clu2.

### 4.2. Sample Collection and Cell Fraction Enrichment

Bone marrow (BM) samples for molecular studies were obtained during standard diagnostic procedures. Mononuclear BM cells were obtained by Ficoll-Hypaque density gradient centrifugation. An immunomagnetic beads-based strategy (MACS system, Miltenyi Biotec, Auburn, CA) was employed to isolate both plasma cells and progenitor populations. More specifically, CD138+ cells were purified by positive selection with a specific anti-CD138 antibody; in 22 patients, the B-cell fraction was enriched by depletion of all non-B cells (T cells, NK cells, monocytes, dendritic cells, granulocytes, platelets, and erythroid cells) using a cocktail of biotinylated CD2, CD14, CD16, CD36, CD43, and CD235a antibodies. The purity of positively selected plasma cells was assessed by flow cytometry using CD138, CD38, and CD45 antibodies; similarly, the CD138 negative fraction was evaluated before and after separation for the presence of CD19 and CD27 markers (Miltenyi Biotech, Auburn, CA).

### 4.3. RNA Processing and Droplet Digital PCR (ddPCR)

RNA was isolated using the Maxwell^®^ 16 Total RNA purification kit and then at least 100 ng was processed with the SuperScript^®^ IV Reverse Transcriptase to obtain the cDNA. Next, 20 μL reaction mixtures containing the 2x ddPCR Supermix for probes (no dUTP), 20x target primers/probe mix (FAM/HEX), cDNA templates were prepared and loaded into the QX200™ Droplet generator. After droplet generation, droplets were carefully transferred into a 96-well plate and a thermal cycler was performed as follows: 95 °C for 10 min for enzyme activation, denaturation at 94 °C per 30 s, and annealing/extension at 58 °C for 1 min, repeated for 40 cycles, and, finally, enzyme deactivation for 10 min. Ramp rate was set at 2 °C/s and annealing temperature was maintained at 58 °C for all the designed assays. Droplet generation and transfer of emulsified samples to PCR plates were performed according to the manufacturer’s instructions (QX200™ Droplet Digital PCR—ddPCR™ System—Bio-Rad; Instruction manual, QX200™ Droplet Generator—Bio-Rad, Hercules, CA, USA).

### 4.4. Data and Statistical Analysis

Gene expression data derived from microarray were normalized and processed as previously described [11]. Briefly, in order to obtain model-based expression values, all samples were normalized and analyzed by invariant set normalization using dChip software [29], where a baseline array was automatically selected according to its median brightness and good call percentage among the array group (MM_107); data were modeled, and perfect match intensities were background adjusted. In some cases, several probes in the array were annotated on the same gene. Pearson’s correlation coefficient was evaluated between these multiple probes over all samples, and only those with the highest coefficients were selected. The absolute quantity of DNA per sample (copies/μL) was evaluated by QuantaSoft™ Software (v.1.7.4). Comparisons between patients’ groups were performed using Pearson’s χ2 test or Fisher’s exact test, as appropriate, for categorical data, and the Kolmogorov–Smirnov test for continuous data. The %CV (Standard deviation/Mean*100) was also used to assess variability.

## Figures and Tables

**Figure 1 ijms-23-12450-f001:**
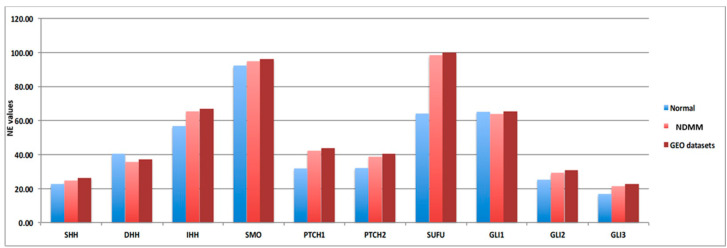
Hedgehog genes are low abundant targets in different NDMM patients’ cohorts. Normalized expression (NE) values of the ten Hedgehog (HH) genes included in the signature are reported in y-axis related to the normal donor (light blue bar—27 samples), our NDMM cohort (light red bar—126 samples) and GEO dataset (dark red bar—3023 samples).

**Figure 2 ijms-23-12450-f002:**
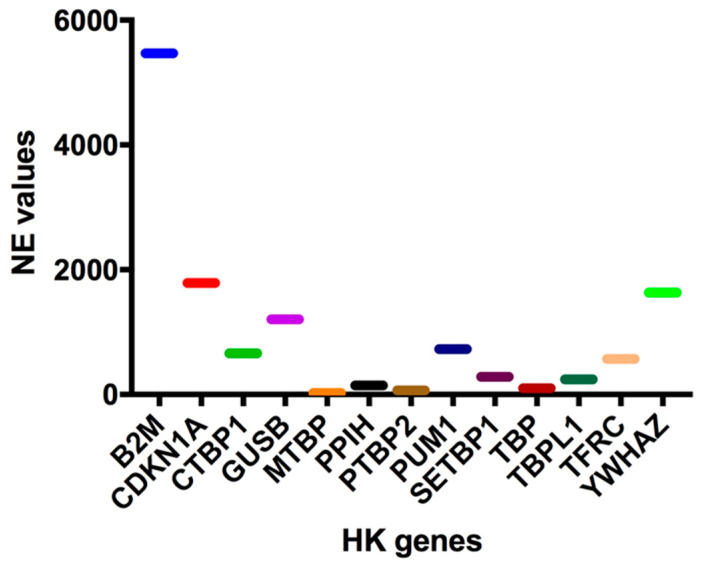
Housekeeping Genes (HK) normalized expression (NE) values. In order to normalize the 10-HH genes quantification values, expression data have been normalized using a housekeeping gene. Therefore, the normalized expression (NE) values of at least 13 known housekeeping genes (B2M, CDKN1A, CTBP1, GUSB, MTBP, PPIH, PTBP2, PUM1, SETBP1, TBP, TBPL1, TFRC, YWHAZ) were extrapolated from our 123-gene expression array data.

**Figure 3 ijms-23-12450-f003:**
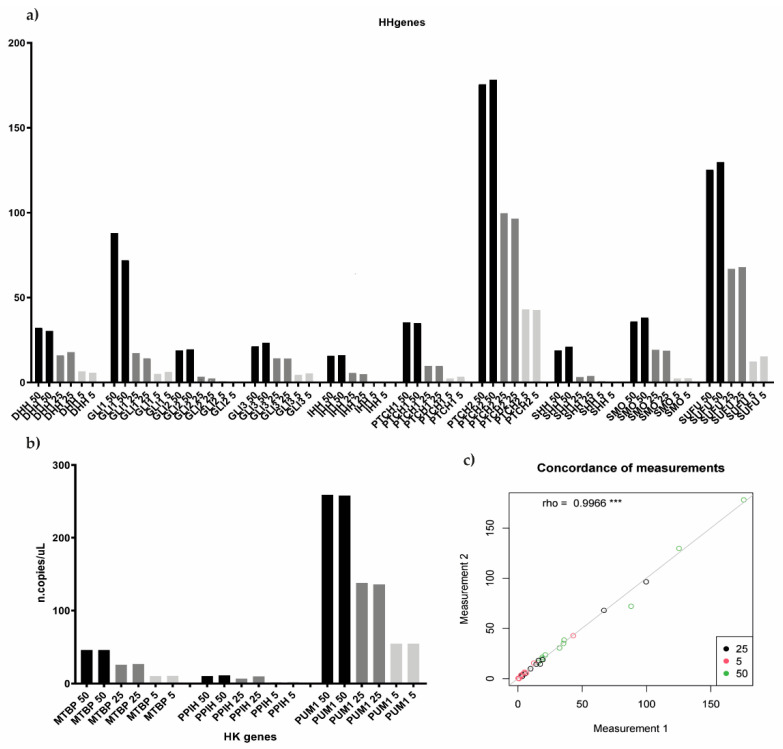
Testing different target input concentration for both Hedgehog and housekeeping genes. (**a**,**b**) histograms showed the reproducibility between both different experiments and different replicates, 5, 25, and 50 ng of input material were tested in replicates, two times in two independent experiments, both for HH as well as for housekeeping genes. (**c**) showed the level of concordance between the different samples’ measurement according to the different target input concentration tested (rho = 0.9966). *** *p* < 0.05.

**Figure 4 ijms-23-12450-f004:**
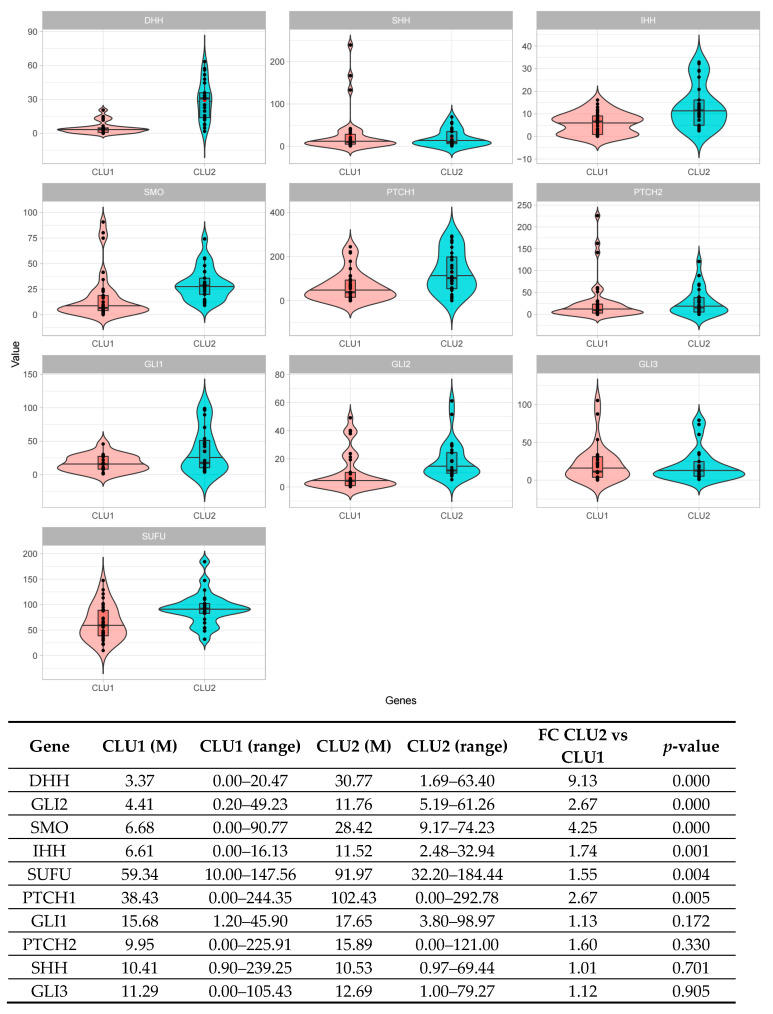
Digital PCR validation of the 10 HH genes signature across the retrospective NDMM patients’ cohort. The comparison between patients’ subgroups (Cluster 1—CLU1 and Cluster 2—CLU2) stratified according to the 10 HH gene signature effectively results in an upregulation of the pathway in Cluster 2 compared to Cluster 1, as detected by the digital PCR assays (6/10 genes *p* < 0.05). Genes are illustrated according to their role in the pathway: SHH, DHH, and IHH (the ligands); PTCH1, PTCH2, and SMO (the receptors); SUFU (pathways’ intermediate); and GLI1, GLI2, and GLI3 (the transcription factors).

**Table 1 ijms-23-12450-t001:** Hedgehog gene expression values comparison between 9 different NDMM patients’ cohorts. The normalized expression values of 10 HH genes were compared between 9 different cohorts of NDMM patients downloaded from GEO database (https://www.ncbi.nlm.nih.gov/geo/, accessed on 1 July 2022) for a total of 3023 patients versus 27 donor samples in order to determine their abundancy in the context of myeloma disease (M: average values; FC: fold change).

PROBE SET	GENES	DONOR (M)	MM (M)	FC MM vs. DONOR	*p*-Value
207586_at	SHH	22.77	23.1	0.32	7.42 × 10^−7^
1552730_at	DHH	40.39	40.4	−0.01	8.60 × 10^−1^
229358_at	IHH	56.7	56.98	−0.27	8.65 × 10^−5^
218629_at	SMO	92.18	92.33	−0.15	1.41 × 10^−2^
209816_at	PTCH1	31.76	32.32	−0.56	1.43 × 10^−14^
221292_at	PTCH2	32.08	32.28	−0.2	4.95 × 10^−4^
222749_at	SUFU	64.04	64.46	−0.42	2.16 × 10^−8^
206646_at	GLI1	65.18	65.55	−0.37	2.77 × 10^−8^
207034_at	GLI2	25.31	25.8	−0.49	2.22 × 10^−12^
1569342_at	GLI3	16.7	16.81	0.11	2.32 × 10^−1^

**Table 2 ijms-23-12450-t002:** Testing the 10 HH genes signature in MM cell lines. The signature was able to classify each cell line as belonging either to Cluster 1 (MM1S and RPMI-8226) or to Cluster 2 (KMS12.BM and NCI-H929), according to their distance in terms of gene expression from CLU1 or CLU2.

Sample ID	CLU1	CLU2	Classification
GSM562812_KMS12BM	18.7636	17.1984	2
GSM562817_NCIH929	20.0729	15.7932	2
GSM1300960_MM1S	10.3441	17.6051	1
GSM1300964_RPMI8226	7.40587	13.3582	1
GSM562814_MM1S	10.3498	10.4041	1
GSM562819_RPMI8226	11.3982	14.13	1
GSM1300957_KMS12	15.2754	10.3545	2
GSM1300962_NCIH929	15.1557	13.777	2

**Table 3 ijms-23-12450-t003:** ddPCR multiplexed assay validation of a 10 HH gene signature. The different assays have been multiplexed in order to optimize the cartridge by coupling genes with similar values of expression in the same well. SMO, SHH, PTCH1, IHH, DHH, and MTBP were conjugated with FAM fluorophores, whereas SUFU, GLI1, GLI2, GLI3, and PTCH2 were conjugated with HEX fluorophores.

10-HH Genes SIGNATURE ddPCR Assay	
FAM	HEX
*IHH*	*GLI2*
*PTCH1*	*GLI1*
*SMO*	*GLI3*
*DHH*	*SUFU*
*SHH*	*PTCH2*

**Table 4 ijms-23-12450-t004:** Digital PCR validation of the 10 HH genes signature across the prospective NDMM patients’ cohort. A prospective cohort of 22 patients was employed to validate the ddPCR assay on CD138+ plasma cells. An immunophenotypic characterization of the neoplastic clone permit the stratification according to their differentiation status. In particular, 10 out of 22 were CD19+/CD81+ (CLU1—immature/less differentiated phenotype) and 12 out of 22 were CD19-/CD81- (CLU2—mature/more differentiated phenotype). Screening of these patients with the 10 HH-genes signature confirms their differentiation status. Indeed, the comparison of the absolute quantification values between the two groups highlights the overexpression of 10 HH genes signature in CD19-/CD81- patients, the most differentiated group.

Gene	CD19-/CD81- (M)	CD19-/CD81- (Range)	CD19+/CD81+ (M)	CD19+/CD81+ (Range)	FC CD19-/CD81- vs. CD19+/CD81+	*p*-Value
GLI1	108.28	2.50–413.17	5.8	3.29–81.47	18.67	0.009
PTCH1	174.65	30.46–2363.64	63.29	5.80–135.74	2.76	0.028
SMO	296.26	136.36–1045.45	94.2	6.70–1464.60	3.15	0.047
IHH	172.15	18.56–1056.82	100.8	2.90–1542.49	1.71	0.118
PTCH2	321.45	75.0–1315.91	172.66	27.12–1695.13	1.86	0.136
SHH	165.71	11.17–1213.64	94.2	3.68–1458.37	1.76	0.177
GLI3	212.5	19.90–387.60	123.29	14.50–527.53	1.72	0.201
GLI2	242.39	147.15–1343.18	220.4	17.99–1445.91	1.10	0.256
DHH	82.4	9.14–422.75	35.66	−5.8–186.36	2.31	0.394
SUFU	183.55	44.88–502.99	177.8	50.13–396.16	1.03	0.887

## Data Availability

Not applicable.

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
