# Peer review of "Identification of a Maturation Plasma Cell Index through a Highly Sensitive Droplet Digital PCR Assay Gene Expression Signature Validation in Newly Diagnosed Multiple Myeloma Patients"

_ijms, 2022, doi:10.3390/ijms232012450_

Round 1
Reviewer 1 Report
Overall, an interesting study to stratify multiple myeloma patient samples using a ddPCR signature. This represents a definite improvement over microarray or real-time PCR technologies. The authors spent considerable time validating their gene panel. One drawback is that the ddPCR technique described is limited to 10 hedgehog genes but this could be easily expanded.
line 93 - reference 11 isn't properly formatted
Author Response
Dear Reviewer 1,
thanks for your review and I appreciate your comments.
Our work has beeen focused only on 10 genes, but it can be expanded to several other genes, as you suggested. Our limited gene selection has been centered on a Hedgehog pathway that is fundamental for self-renewal in plasma cells and precursor states and might justify the cell maturation state. Our future studies will address deepenly the integration of gene and immunophenotype expression by integrating more informative markers.
Reviewer 2 Report
The authors used highly sensitive droplet digital PCR assay to identify gene expression signature in newly diagnosed myeloma plasma cell. It is interesting and with potential for further survey of plasma cell signature in the future.
Author Response
Dear Reviewer 2,
thanks a lot for your review and your comments.
Following your suggestions, we have implemented bibliography with the latest publications around this topic and expand the introduction accordingly. In addition, we have revised both methods and also results sections in order to make sure to eliminate approximations or unclear concepts.
Reviewer 3 Report
Droplet digital PCR is a novel, highly sensitive method that is automatized and relatively easy to interpret. Authors of this interesting manuscript applied this novel technique to establish a maturation plasma cell index through profiling a 10-member Hedgehog gene signature panel. The assay was validatated on a considerably large primary myeloma patient plasma cell cohort. It was shown that this relatively easy procedure can reproduce the gene expression profiling data quite well.
The data presented are fairly convincing, nevertheless the fact that with this relatively straightforward method only 22 NDMM patient samples were prospectively analized is somewhat disappointing. Additionally, while important surface marker expression data are presented descriptively, only speculations are given with respect to the clinical behavior of these patients. Therefore, further clinical relavance of such a maturation index remains murky at this point.
Author Response
Dear Reviewer 3,
we kindly appreciate your review and your comments.
Our first aim was the validation of microarray data with ddPCR data and this has been done successfully on a large and retrospective cohort of primary samples. I agree with you that the prospective validation is on a limited number of patients with respect to the retrospective analysis, but I would like to emphasize the relevance of these data even if in a small patients' cohort, since to obtain two well defined cohorts of patients (11 patients with immature CD19+/CD81+ plasma cells, and 11 patients with mature CD19-/CD81- plasma cells), each of them with both plasma cells and B cells fractions sufficiently pure and populated in order to be used for downstream analyses, is extremely difficult. For this reason, more than 60 patients have been screened in order to obtain these results. Since these are only preliminary data from a clinical point of view, we are currently working on the clinical validation in a larger homogenous treated cohort of newly diagnosed multiple myeloma patients.